# Exploiting Task Relationships for Continual Learning with Transferability-aware Task Embedding

## Abstract

Continual learning (CL) has been a crucial topic in contemporary deep neural network usages, where catastrophic forgetting (CF) can impede a model's ability to progressively acquire knowledge, leading to critical training inefficiency and constraint in the improvement of model's overall capacity. Existing CL strategies mostly mitigate CF either by regularizing model weights and outputs during finetuning or by distinguishing task-specific and task-sharing model components to adapt the training process accordingly. Yet despite their effectiveness, these previous explorations are mainly limited to elements of task models, while we speculate a deeper exploitation of interrelationship among tasks can provide more enhancement for CL. Therefore, to better capture and utilize the task relations, we propose a transferability task embedding guided hypernet for continual learning. By introducing the information theoretical transferability based task embedding named H-embedding and incorporating it in a hypernetwork, we establish an online framework capable of capturing the statistical relations among the CL tasks and leveraging these knowledge for deriving task-conditioned model weights. The framework is also characterized by notable practicality, in that it only requires storing a low dimensional task embedding for each task, and can be efficiently trained in an end-to-end way. Extensive evaluations and experimental analyses on datasets including Permuted MNIST, Cifar10/100 and ImageNet-R showcase that our framework performs prominently compared to various baseline methods, as well as displays great potential in obtaining intrinsic task relationships.

## 1 Introduction

Continual learning (CL), also known as incremental learning or life-long learning, has been an essential topic in the modern application of deep neural networks, where a model is expected to learn a series of tasks sequentially for the optimization of its capability (Wang et al., 2024). However, in practical usages, catastrophic forgetting (CF) (Kirkpatrick et al., 2017) can hamper the model from accumulatively gaining knowledge as intended, severely hindering the overall growth of model capacity and resulting in significant wastes of training resources. Specifically, in CL settings, a model is trained one-by-one (*i.e.* data in the old tasks are not fully available anymore when training new ones) on a sequence of tasks, which typically contains either category change or data distribution shifts (Qu et al., 2021). Consider the training process of a new task, a desirable model performance should be characterized by two aspects (von Oswald et al., 2020). 1) **Backward transfer / non catastrophic forgetting** (Kirkpatrick et al., 2017)**:** improvement or at least no significant degradation on previous tasks. 2) **Forward transfer:** higher efficiency in learning the new task compared to training a model from scratch.

Up to now, there have been many studies dedicated to CL strategies, with most of them involving rehearsal of previous data to alleviate the knowledge degradation caused by CF. However, the growing concerns of privacy and data safety have made this solution not always feasible, bringing increased attention to the rehearsal-free CL setting (Smith et al., 2023, more discussion in Sec. 2). Previous works on rehearsal-free CL can be mostly described as regularization-based approaches, including parameter space regularization (e.g. Kirkpatrick et al., 2017; Zenke et al., 2017) and feature space regularization (e.g. Li & Hoiem, 2017; Rebuffi et al., 2017), with the latter alternatively referred to

as distillation approaches. These approaches introduce different regularization losses during training on the model weights, the intermediate layer or the final output. Despite their effectiveness, regularization methods are limited in their design that the models learned on subsequent tasks are forced to stay close to those on previous tasks, while in fact the tasks in a CL setting are not necessarily similar or relevant to each other.

This brings us to a fundamental question in continual learning: *Having positive backward and forward transfer in CL relies on capturing the underlying relationship between different tasks, but how do we learn and utilize such relationship as we train the model?* Architecture-based approaches (Wang et al., 2024) have recently emerged as a tentative answer to this question[1]. Rather than attempting to forcibly align all tasks in their model parameters or outputs, they generally aim at dedicating task-specific and task-sharing model components from different architecture levels (e.g. Mallya & Lazebnik, 2018; Wortsman et al., 2020; Jin & Kim, 2022), and tailoring the training process accordingly. Nevertheless, the allocation of model parts to different tasks usually comes with scaling problems with the growth of task number, potentially resulting in either insufficient model capacity or excessive model size growth. On the other hand, as a variant of these approaches, von Oswald et al. (2020) introduces a task-conditioned hypernet, allowing for comprehensive generations of task-specific model weights based on the corresponding task embeddings without trying to draw a clear distinction between common and specific model elements. However, the original hypernet relies solely on the black-box learning of task embeddings to compute task relationships. This approach can be inefficient in capturing the true relationship between tasks in the high dimensional task space, as it overlooks the prior knowledge that can be incorporated using statistical tools.

Therefore, with the aim of a better understanding and utilization of the task space in CL, we propose in this work a novel hypernet framework, where the learning of task embedding is guided by dynamically estimated task relationships. As transferability scores (Ding et al., 2024, see Sec. 2) aims to statistically estimate the fitness of source models on target tasks, it can provide prior information on the relationship between new tasks and old tasks. Based on these desiderata, we optimize the hypernet to learn task embeddings that preserve the transferability between the current task and prior tasks. Particularly, we propose an online task embedding scheme named H-embedding, which distills the transferability information into the hypernet through an optimization process. H-embedding can be learned efficiently without accessing previous data by maximizing the consistency between their Euclidean distances and the H-score (Bao et al., 2019) transferability among the corresponding tasks. To match the magnitude of the task embedding distance with that of the transferability score, we further introduce a learnable scaling constant in H-embedding. This significantly enhances the learning stability over long task sequence without sacrificing performance.

Our H-embedding can be seamlessly incorporated into the hypernet via an encoder-decoder module to ensure its alignment with the learned task embedding[2], serving as a guidance for the training of hypernetwork. With the introduction of H-embedding guidance, the framework has markedly improved in its capability of capturing task relationship, featured by: 1) an efficient and reliable learning of task embedding based on the information theoretical foundation of H-score metric; 2) a notable enhancement of CL in its forward transfer performance; 3) ease of practical use with end-to-end training and no extra storage required beyond the low-dimensional task embeddings.

## 2 RELATED WORKS

### 2.1 REHEARSAL-FREE CONTINUAL LEARNING

The strictness of CL settings varies with the extent of allowed previous data accessibility. Multi-task learning (Caruana, 1997), with full data availability of all tasks, can actually be viewed as a special case and upper-bound of CL, while rehearsal-free CL (Smith et al., 2023), with non previous data involved in the training of new tasks, is on the other hand the strictest CL setting under this criteria. In spite of the success of rehearsal based methods in various benchmarks (Bang et al., 2021; Shin et al., 2017; Belouadah & Popescu, 2019), rehearsal-free CL is catching the attention of researchers

---

[1]A large proportion of these approaches actually require certain access to previous data and hence do not conform to our rehearsal-free setting. We cover them here mainly for a comprehensiveness of discussion.

[2]In fact, this guidance is more general and can be incorporated into any hypernet-based CL framework, yet here we mainly base our framework on the work of von Oswald et al. (2020).

recently (Smith et al., 2023) because of its low dependency on revisiting previous tasks and therefore broader application in the era of growing data privacy concern.

Existing works on rehearsal-free CL are generally based on regularization strategies. EWC (Kirkpatrick et al., 2017) and SI (Zenke et al., 2017) introduce penalties to restrict the alteration of parameters vital for addressing prior tasks, thereby reducing the risk of catastrophic forgetting. LwF (Li & Hoiem, 2017; Rebuffi et al., 2017) proposes a cross entropy loss between the predicted class distribution of the *(n-1)*-th task, as generated by the model before and after learning the *n*-th task. Smith et al. (2023) gives an overview of these methods and proposes regularization combinations for better CL performance. In this work, we follow these works and focus on the more challenging rehearsal-free CL setting.

## 2.2 HYPERNETS

Hypernets (Ha et al., 2017), or hypernetworks, are specialized neural networks that produce weights for another neural network, *i.e.*, the target network. Recently, they have gained recognition as a potent tool in deep learning, providing advantages such as increased flexibility, adaptability, dynamic nature, training efficiency and model compression (Chauhan et al., 2023). Hypernets have yielded encouraging results in various deep learning applications, including continual learning (von Oswald et al., 2020), causal inference (Chauhan et al., 2024), domain adaptation (Volk et al., 2022), uncertainty quantification (Krueger et al., 2017), few-shot learning (Sendera et al., 2023), and reinforcement learning (Sarafian et al., 2021).

## 2.3 TRANSFERABILITY METRICS

Task transferability (Zamir et al., 2018) investigates the relationships between tasks and provides an effective method for evaluating and selecting source tasks in transfer learning. It also plays a crucial role in developing strategies for multi-task learning and meta-learning. For the ease of usage, previous studies have proposed metrics based on task models and data distributions for a quick estimation of transferability (Ding et al., 2024). H-score (Bao et al., 2019; Ibrahim et al., 2022; Wu et al., 2024) uses an information-theoretic framework to evaluate transferability by solving a maximum correlation problem. NCE (Tran et al., 2019) employs conditional entropy to assess transferability and task difficulty. LEEP score (Nguyen et al., 2020; Agostinelli et al., 2022) offers a more generalized metric, defined by measuring the performance of a classifier developed from source model predictions when applied to the target task. LogME (You et al., 2021) assesses target task accuracy using a formulation integrating all possible linear classifiers derived from source model features. OTCE (Tan et al., 2021; 2024) combines optimal transport with conditional entropy to both estimate the domain and task difference between source and target. These metrics are mostly designed with differed assumptions and source accessibility, with their use applicable to different problem settings.

# 3 PRELIMIMARY

## 3.1 MATHEMATICAL FORMULATION

Consider a problem setting consisting of $M$ tasks $\{T_j\}_{j=1}^M$, the data of task $j$ is denoted by $D_j = (\mathbf{X}^{(j)}, \mathbf{Y}^{(j)})$, with input samples $\mathbf{X}^{(j)} = \{\mathbf{x}^{(j,i)}\}_{i=1}^{N_j}$ and output samples $\mathbf{Y}^{(j)} = \{\mathbf{y}^{(j,i)}\}_{i=1}^{N_j}$. Here, $N_j = |\mathbf{X}^{(j)}| = |\mathbf{Y}^{(j)}|$ denotes the sample size of the $j$-th task, and the attributes of sample data $\mathbf{x}^{(j,i)}, \mathbf{y}^{(j,i)}$ depends on the particular CL setting as well as the form of tasks. In CL, the $M$ tasks are learned sequentially during the training stage. To be specific, denoting a neural network model as $f(x, \Theta)$ (where $f$ represents the model function, $x$ represents the input data, and $\Theta$ represents the model weights) and the model weights acquired in task $j-1$ as $\Theta^{(j-1)}$, the goal of learning task $j$ is to derive a new set of weights $\Theta^{(j)}$ that not only achieves the optimal performance on task $j$, but also performs better or not significantly worse than $\Theta^{(j-1)}$ on tasks $T_1, \ldots, T_{j-1}$. For a rehearsal-free CL setting, the previous data $D_1, \ldots, D_{j-1}$ are not accessible during the training of the $j$-th task.

Based on the discrepancy between $D_{j-1}$ and $D_j$, Hsu et al. (2018) and Van de Ven & Tolias (2019) categorized CL settings into three specific scenarios: task incremental, class incremental, and domain incremental. Table 1 summarizes the differences among these scenarios. For a better con-

centration on the study of CL methodology, our work only focuses on the task incremental CL. In this scenario, the output spaces of tasks are partitioned by task IDs and are mutually exclusive between $D_{j-1}$ and $D_j$, which is denoted as $\mathbf{Y}^{(j-1)} \neq \mathbf{Y}^{(j)}$. It can be then naturally indicated that $P(\mathbf{Y}^{(j-1)}) \neq P(\mathbf{Y}^{(j)})$ and $P(\mathbf{X}^{(j-1)}) \neq P(\mathbf{X}^{(j)})$. Notably, here task IDs are accessible during both training and testing phases.

| Scenario | $P(\mathbf{X}^{(j-1)}) \neq P(\mathbf{X}^{(j)})$ | $P(\mathbf{Y}^{(j-1)}) \neq P(\mathbf{Y}^{(j)})$ | $\mathbf{Y}^{(j-1)} \neq \mathbf{Y}^{(j)}$ | Task ID |
|---|---|---|---|---|
| Domain Incremental | ✓ | ✗ | ✗ | ✗ |
| Class Incremental | ✓ | ✓ | ✗ | ✗ |
| Task Incremental* | ✓ | ✓ | ✓ | ✓ |

Table 1: **Categorization of CL settings based on the discrepancy between $D_{j-1}$ and $D_j$.** '*' denotes the scenario focused on in our work.

## 3.2 H-SCORE

H-score is firstly introduced by Huang et al. in 2019 as a metric assessing the informativeness of features for a task. Theoretically derived from the maximal correlation interpretation of deep neural networks, its mathematical foundation roots to the information theory work known as maximal correlation analysis, which originates from the works of Hirschfeld, Gebelein and Renyi (Hirschfeld, 1935; Gebelein, 1941; Rényi, 1959) and has been followed and further explored by a broad spectrum of successive work. The H-score of $\boldsymbol{f}$ with regard to the task casting $X$ to $Y$ is defined as:

$$H(\boldsymbol{f}) = tr(cov(\boldsymbol{f}(X))^{-1}cov(\mathbb{E}_{P_{X|Y}}[\boldsymbol{f}(X)|Y])). \tag{1}$$

Subsequent work has extended H-score to also serve as a metric for transferability and validated its efficiency with extensive experiments (Bao et al., 2019; Ibrahim et al., 2022), implying the potential of H-score for transfer learning and its application in related problems. With input data $X$, label $Y$ and feature extractor function $\boldsymbol{f}(X)$. The choice of H-score employment in our framework is because of its strong theoretical reliability, conformity of assumption to our problem setting, as well as its non-dependence on source data which makes possible an online embedding estimation.

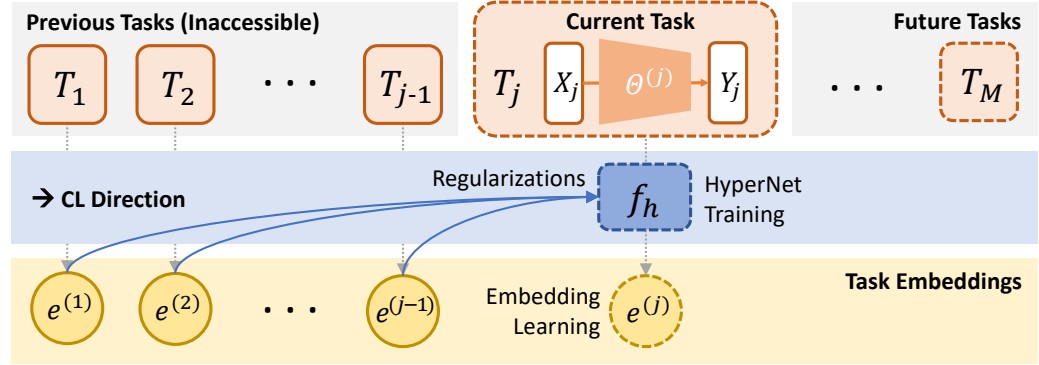

Figure 1: **Illustration of the CL status on the step of learning task $j$ under our framework.** The hypernet is being trained to provide the optimal task model weight $\Theta^{(j)}$ concurrently with the learning of current task embedding $e^{(j)}$, where regularizations are applied using previous embeddings.

## 4 METHODOLOGY

### 4.1 EMBEDDING GUIDED HYPERNET FRAMEWORK

As mentioned in the Introduction, unlike most existing approaches that constrain the magnitude or range of variations in model weights $\Theta$ or outputs $f(x, \Theta)$ during the training of task $j$ to achieve

maximum proximity between $\{\Theta^{(j)}; f(x, \Theta^{(j)})\}$ and $\{\Theta^{(j-1)}; f(x, \Theta^{(j-1)})\}$, we address the CL problem from a meta perspective. In this work, a hypernet framework is proposed to capture the underlying relation between different tasks and maximally leverage our prior knowledge about task interrelationships.

Following von Oswald et al. (2020), we introduce a task-conditioned hypernetwork $f_h(e, \Theta_h)$ with hypernet weights $\Theta_h$ to map a task embedding $e$ to its corresponding model weights $\Theta$, and present a framework that guides the hypernet with transferability based task embeddings $\hat{e}$. Specifically, all tasks $\{T_j\}_{j=1}^{M}$ in the learning scenario share a single hypernet $f_h$ that generates their task model weights using their task-specific embeddings $\{e^{(j)}\}_{j=1}^{M}$, i.e. $\Theta^{(j)} = f_h(e^{(j)}, \Theta_h)$ for task $j$. During each task $T_j$ in training, the task embedding $e^{(j)}$ is learned via gradient descent simultaneously with the updating of hypernet parameters $\Theta_h$, while parameters other than $\Theta_h$ and $e^{(j)}$ are fixed and can be viewed as constants. The loss to be minimized is composed of three parts:

- The target loss, a supervised loss to learn the current task $j$.

$$L_t = \mathcal{L}(f(x^{(j)}, \Theta^{(j)}), y^{(j)}) = \mathcal{L}(f(x^{(j)}, f_h(e^{(j)}, \Theta_h)), y^{(j)}) \tag{2}$$

- The continual learning loss (same as introduced by von Oswald et al. (2020)), to prevent CF by ensuring that given previous task embeddings $\{e^{(n)}\}_{n=1}^{j-1}$, the network weights output by the hypernet before and after training on task $j$ are analogous.

$$L_c = \frac{1}{j-1} \sum_{n=1}^{j-1} L_c^{(n)} = \frac{1}{j-1} \sum_{n=1}^{j-1} ||f_h(e^{(n)}, \Theta_h) - f_h(e^{(n)}, \Theta_h^*)||^2 \tag{3}$$

- The embedding regularization loss to provide the hypernet with additional prior knowledge about the task relationships.

$$L_e = L_e(e^{(j)}, \hat{e}^{(j)}) \tag{4}$$

Here, $\mathcal{L}$ denotes certain supervised task loss (cross-entropy loss in our experiments), and $\Theta_h^*$ is the set of hypernet parameters before learning task $j$. The exact form and derivation of the embedding regularization loss $L_e$ will be covered in Sec. 4.2. To summarize, our final loss function is as follows:

$$L = L_t + \beta_e L_e + \beta_c L_c \tag{5}$$

Take a slice on the $j$-th task, our approach for the training of $T_j$ is depicted in Fig. 2. Notably, although it may appear that the task model weights are first generated and subsequently used for inference, the framework is actually end-to-end, with the hypernet parameters $\Theta_h$ and embeddings $e^{(j)}$ optimized directly by feeding the task data and minimizing the total loss. In other words, there is no additional training procedure introduced in our framework, and the only information to save is the task embeddings $e^{(j)}$ (typically of low dimensions[3]). For a more comprehensive view of our guided hypernet framework, we further present the full continual learning procedure in Fig. 1.

## 4.2 EMBEDDING REGULARIZATION VIA ENCODER AND DECODER

As a key contribution of our framework, we introduce an embedding regularization module to incorporate prior information - specifically, the relationships among CL tasks - into the hypernet. The hypernet is composed of an encoder and a subsequent network. During the forward pass, the task embedding $e$ is mapped from the embedding space $\mathcal{E}$ to a hidden feature $h$ in the hidden space $\mathcal{H}$ by the encoder, and then to the weight space $\mathcal{W}$ by the subsequent network. For task $j$, we have:

$$\Theta^{(j)} = f_{h'}(h^{(j)}) = f_{h'}\left(f_{Enc}(e^{(j)})\right) = f_h(e^{(j)}). \tag{6}$$

Here, $f_{Enc}$ and $f_{h'}$ denote the encoder and the rest part of hypernet respectively. From an information transmission perspective, we presume that the hypernet should in its hidden space encode sufficient information to recover the H-embedding $\hat{e}$ that indicates the known relationships among tasks. Therefore, we additionally introduce a trainable decoder to map the hidden feature $h$ to a

---

[3]The dimension of task embedding is set to 32 in Cifar10/100 & ImageNet-R and 24 in MNIST experiments.

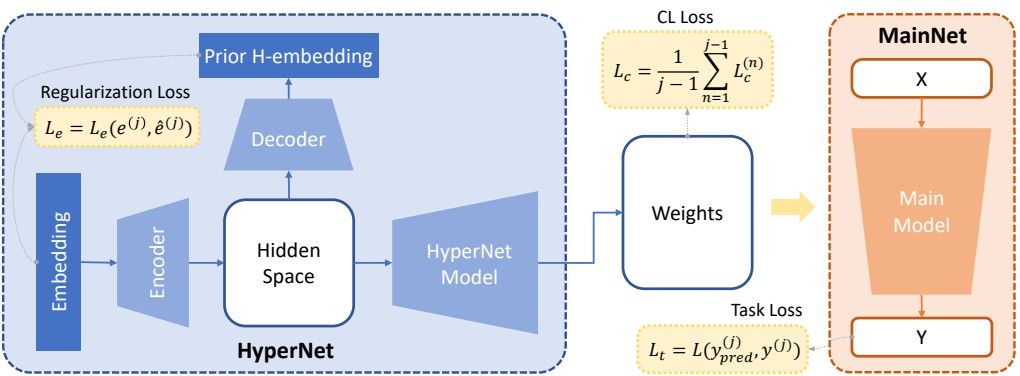

Figure 2: **Framework of our hypernet on the slice of task** $j$. A hypernet (left, blue) is utilized to learn the weights of the main model (right, orange), where the H-embedding guidance is introduced using an encoder-decoder module. The entire framework is trained end-to-end by inputting task data into the main model and propagating gradients backward to update both hypernet and embedding.

embedding $\tilde{e}$ such that the discrepancy between $\tilde{e}$ and the H-embedding $\hat{e}$ should be minimized, *i.e.*, for task $j$

$$\tilde{e}^{(j)} = f_{Dec}(h^{(j)}) = f_{Dec}\left(f_{Enc}(e^{(j)})\right) \tag{7}$$

should be as close to $\hat{e}^{(j)}$ as possible, where $f_{Dec}$ denotes the decoder. Summarize it up in a mathematical form, we have the embedding regularization loss for task $j$:

$$L_e = L_e(e^{(j)}, \hat{e}^{(j)}) = \mathcal{L}\left(f_{Dec}(f_{Enc}(e^{(j)})), \hat{e}^{(j)}\right). \tag{8}$$

$\mathcal{L}$ denotes certain similarity loss, set to the cosine similarity loss in our experiments. The decoder is updated together with the hypernet during training. Notably, the encoder and decoder are both shallow fully connected neural networks, and hence no significant computing cost is posed with the introduction of our embedding regularization module. The influence of this module will be discussed through experimental studies in later sections.

### 4.3 H-SCORE TASK EMBEDDING

In order to guide the hypernet and embedding through the encoder-decoder module, we need to incorporate the interrelationships among tasks implied by the accessible data into a prior embedding. In this work, we propose a H-score based online embedding named H-embedding for the CL tasks.

Particularly, during the training stage of task $j$, we first measure the H-score transferability to $T_j$ from each previous tasks $\{T_n\}_{n=1}^{j-1}$ with $D_j$ and previous task embeddings $\{e^{(n)}\}_{n=1}^{j-1}$, leveraging that the previous task models can be conveniently reconstructed by the hypernet and corresponding task embeddings.

$$H(T_n, T_j) = tr\left(cov(f_l(x^{(j)}, \Theta^{(n)}))^{-1}cov(\mathbb{E}_{P_{X|Y}}[f_l(x^{(j)}, \Theta^{(n)})|y^{(j)}])\right) \tag{9}$$

$$\Theta^{(n)} = f_h(e^{(n)}, \Theta_h) \tag{10}$$

Here, $f_l(*)$ denotes the output of the last layer before the classifier in the main net $f$, which can be viewed as the feature of task data $X^{(j)}$. The H-embedding $\hat{e}^{(j)}$ is then computed by minimizing the difference between the Euclidean distance of $e^{(n)}, \hat{e}^{(j)}$ and their H-score transferability $H(T_n, T_j)$:

$$\hat{e}^{(j)} = \arg\min_{\hat{e}^{(j)}} \sum_{n=1}^{j-1} \left(||\hat{e}^{(j)} - e^{(n)}||_2 - H(T_n, T_j)\right)^2, \tag{11}$$

where $e^{(n)}$ is calculated and stored when learning previous tasks. Consider that the transferability can only be derived with at lease two tasks, the H-embeddings and embedding loss are only computed after the first two tasks. Nevertheless, as a target centered transferability metric, the H-score

may not consistently align in magnitude with the embedding initialization when sequentially learning the tasks in CL. Hence, we further introduce a scaling constant $\gamma^{(j)}$ that would be optimized together with $\hat{e}^{(j)}$ but not utilized later, modifying Eqn. 11 to:

$$\hat{e}^{(j)}, \gamma^{(j)} = \operatorname*{arg\,min}_{\hat{e}^{(j)}, \gamma^{(j)}} \sum_{n=1}^{j-1} \left( ||\hat{e}^{(j)} - e^{(n)}||_2 - \gamma^{(j)} H(T_n, T_j) \right)^2. \tag{12}$$

Given that $H(T_n, T_j)$ and $e^{(n)}$ are actually constants, the above optimization problem is a benign bi-variate optimization problem. We could thus apply a gradient descent algorithm to effectively compute the H-embedding $\hat{e}^{(j)}$ for the $j$-th task. As such, the H-embeddings for all tasks during the continual learning can be calculated in an inductive way. We summarize the entire training process of task $j$ in our H-embedding guided hypernet as Algorithm. 1

---

**Algorithm 1:** H-embedding guided Hypernet: Training of Task $j$

---

**Input:** Task data $D_j$, previous task embeddings $\{e^{(n)}\}_{n=1}^{j-1}$, hypernet weights $\Theta_h$
**Parameter:** Learning rate $\lambda$
**Output:** Current task embedding $e^{(j)}$, updated hypernet weights $\Theta_h$
Randomly initialize $e^{(j)}, \hat{e}^{(j)}, \gamma^{(j)}$;
**if** $j > 2$ **then**
    **for** $n \leftarrow 1$ **to** $j-1$ **do**                          // Compute transferability
        $\Theta^{(n)} \leftarrow f_h(e^{(n)}, \Theta_h)$ ;
        $H(T_n, T_j) \leftarrow tr\left(cov(f_l(x^{(j)}, \Theta^{(n)}))^{-1} cov(\mathbb{E}_{P_{X|Y}}[f_l(x^{(j)}, \Theta^{(n)})|y^{(j)}])\right)$     ▷ Eq. 10
    **end**
    $\hat{e}^{(j)}, \gamma^{(j)} \leftarrow \arg\min_{\hat{e}^{(j)}, \gamma^{(j)}} \sum_{n=1}^{j-1} \left( ||\hat{e}^{(j)} - e^{(n)}||_2 - \gamma^{(j)} H(T_n, T_j) \right)^2$        ▷ Eq. 12
**end**
**repeat**                                                      // Train hypernet
    $e^{(j)} \leftarrow e^{(j)} - \lambda \nabla_{e^{(j)}} L$;
    $\Theta_h \leftarrow \Theta_h - \lambda \nabla_{\Theta_h} L$                                         ▷ Eq. 5
**until** *converge*;
**Return** $e^{(j)}, f_h(\,\cdot\,, \Theta_h)$

---

## 5 EXPERIMENTS

### 5.1 EXPERIMENTAL SETTINGS

#### 5.1.1 BENCHMARKS

To comprehensively verify the effectiveness of our framework and further analyse its reliability, we conduct extensive experiments on three CL benchmarks. **PermutedMNIST** (Goodfellow et al., 2013) benchmark is an variant of MNIST (LeCun et al., 1998), forming CL tasks from the original MNIST dataset by applying random permutations to the input image pixels. The permuting procedure can be repeated in experiments to yield a task sequence of needed length, with each task consisting of 70,000 images (60,000 for training and 10,000 for testing) of digits from 0 to 9. **Cifar10/100** is a benchmark composed of 11 ten-way classification tasks, with a full Cifar10 task and a Cifar100 dataset split into ten tasks (Krizhevsky et al., 2009). The model will be firstly trained on the Cifar10 task with 60,000 images (50,000 for training and 10,000 for testing) and then sequentially trained on the ten Cifar100 tasks, each with 6,000 images (5,000 for training and 1,000 for testing). Built on the basis of ImageNet dataset (Deng et al., 2009), **ImageNet-R** (Hendrycks et al., 2021) contains a wide range renditions of ImageNet classes, covering a total of 30,000 images covering 200 ImageNet classes. In the CL benchmark, ImageNet-R is also split into 10 tasks, each with 20 classes and around 3,000 samples (roughly 2,500 for training and 500 for testing)[4].

---

[4]Training specifics and detailed results of our experimental studies are listed in Appendix A.1, with codes available at `https://anonymous.4open.science/r/H-embedding_guided_hypernet/`.

| Method | $\mathcal{AA}$ ($\uparrow$) | $\mathcal{BWT}$ ($\uparrow$) | $\mathcal{FWT}$ ($\uparrow$) |
|---|---|---|---|
| Finetune | $18.32 \pm 0.70$ | $-66.98 \pm 0.45$ | $10.20 \pm 0.55$ |
| Finetune Head | $15.35 \pm 0.11$ | $-70.52 \pm 0.36$ | $9.63 \pm 0.32$ |
| LwF | $33.02 \pm 0.59$ | $-57.22 \pm 0.64$ | $15.87 \pm 0.45$ |
| EWC | $36.15 \pm 1.48$ | $-53.39 \pm 1.69$ | $15.40 \pm 0.42$ |
| L2 | $39.84 \pm 0.67$ | $-50.27 \pm 0.79$ | $16.14 \pm 0.50$ |
| PredKD + FeatKD | $33.03 \pm 0.76$ | $-56.19 \pm 1.23$ | $14.91 \pm 0.85$ |
| PackNet | $71.78 \pm 0.11$ | $0.00$ (N/A) | $1.86 \pm 0.49$ |
| WSN | $82.87 \pm 0.20$ | $0.00$ (N/A) | $13.94 \pm 0.37$ |
| Vanilla Hnet | $82.21 \pm 0.23$ | $-0.05 \pm 0.05$ | $12.82 \pm 0.53$ |
| Rand-embed Hnet | $82.42 \pm 0.17$ | $-0.12 \pm 0.11$ | $12.70 \pm 0.60$ |
| H-embed Hnet* | $83.51 \pm 0.17$ | $-0.06 \pm 0.04$ | $14.25 \pm 0.57$ |

Table 2: **Accuracy (%) Comparison on Cifar10/100.** All range of results are derived by three times of running with different random seeds and calculating the average and standard deviation. Our method (marked by '*') achieves the top average accuracy with high confidence.

### 5.1.2 EVALUATION METRICS

Following our desiderata stated in Sec. 1 as well as the metrics introduced in previous works (Qu et al., 2021; Wang et al., 2024), we evaluate the different CL methods from three aspects:

- **Overall performance**, measured by average accuracy (AA) of final model on all CL tasks:

$$\mathcal{AA} = \frac{1}{M} \sum_{j=1}^{M} a_{j,M} \;;$$

- **Memory degradation of old tasks**, measured by average backward transfer (BWT):

$$\mathcal{BWT} = \frac{1}{M-1} \sum_{j=1}^{M-1} (a_{j,M} - a_{j,j}) \;;$$

- **Learning enhancement of new tasks**, measured by average forward transfer (FWT):

$$\mathcal{FWT} = \frac{1}{M-1} \sum_{j=2}^{M} (a_{j,j} - \tilde{a}_j).$$

Here, $a_{i,j}$ denotes the accuracy (%) measured on the test set of $i$-th task after learning the $j$-th task, and $\tilde{a}_j$ denotes the test accuracy derived by training a randomly initialized model directly on the $j$-th task. To conclude, a most desirable CL strategy should come with higher results on all three metrics, i.e. $\mathcal{AA}$, $\mathcal{BWT}$ and $\mathcal{FWT}$.

### 5.2 PERFORMANCE EVALUATION

### 5.2.1 COMPARISON EXPERIMENTS

Our primary evaluation study is conducted on the Cifar10/100 benchmark with a total of 11 tasks. To ensure fairness in comparison, a non pre-trained ResNet-32 (He et al., 2016) is selected as the backbone model for all the chosen baselines. We reproduce all the methods with our own codes and each methods are run three times with 100 epochs and shared random seeds.

The choice of baselines is based on the requirements that they should both conform to our rehearsal-free setting and be applicable to the benchmark and backbone. For a thorough comparison with existing methods to the greatest extent possible, we selected representative baselines of varied methodology categories, including: **Basic Methods:** Finetune, Finetune Head; **Regularization Methods (Basic):** LwF (Li & Hoiem, 2017), EWC (Kirkpatrick et al., 2017), L2, PredKD+FeatKD (Smith et al., 2023); **Architecture Methods:** PackNet (Mallya & Lazebnik, 2018), WSN (Kang et al., 2022); **HyperNet Methods:** HyperNet (Vanilla Hnet) (von Oswald et al., 2020), HyperNet with random guidance (Rand Guide Hnet). We summarize the experimental results in Table. 2. As can be

| Setting | PermutedMNIST | | | Cifar10/100 | | | ImageNet-R | | |
| | MLP | | | CNN | | | ResNet-32 | | |
| Method | $\mathcal{AA}$ | $\mathcal{BWT}$ | $\mathcal{FWT}$ | $\mathcal{AA}$ | $\mathcal{BWT}$ | $\mathcal{FWT}$ | $\mathcal{AA}$ | $\mathcal{BWT}$ | $\mathcal{FWT}$ |
|---|---|---|---|---|---|---|---|---|---|
| Vanilla Hnet | 97.495 | 0.007 | 0.063 | 69.679 | -7.790 | 7.970 | 38.485 | -0.250 | 7.132 |
| Rand-embed Hnet | 97.464 | **0.019** | 0.021 | 71.179 | **-6.140** | 7.970 | 38.860 | -0.214 | 7.513 |
| H-embed Hnet* | **97.570** | -0.013 | **0.156** | **71.768** | -6.440 | **8.950** | **39.547** | **-0.162** | **8.168** |

Table 3: **Ablation Study on different benchmarks and backbones.** Our H-embedding guidance proves to be effective across all three settings, attaining the highest average accuracy, with competitive backward transfer and the best forward transfer performance.

told from the table, our method perform prominently in the ultimate acquisition of CL tasks, achieving the highest final average accuracy. It also derives the best overall ability in terms of forward and backward transfer, outperforming all architecture based and hypernet baselines in FWT as well as surpassing regularization baselines by a large margin in BWT.

### 5.2.2 ABLATION STUDIES

To broaden the comprehensiveness of evaluation and take on a better concentration on validating our introduction of H-embedding guidance, we conducted extra ablation studies on three differed settings with different benchmarks as well as model backbones. Namely, experimental settings include: PermutedMNIST (10 tasks) using an MLP model, Cifar10/100 (11 tasks) using a 4-layer CNN model, and ImageNet-R (10 tasks) using a ResNet-32 model. The performance are evaluated and summarized in Table. 3, where a broad increase in CL performance could be observed across all benchmarks and backbones.

### 5.3 DISCUSSION AND IN-DEPTH PERFORMANCE ANALYSIS

For a better analysis of the effectiveness of our strategy, we further investigate the detailed training behaviour displayed in different CL strategies, showing that our H-embedding guided hypernet is characterized by the following superiority.

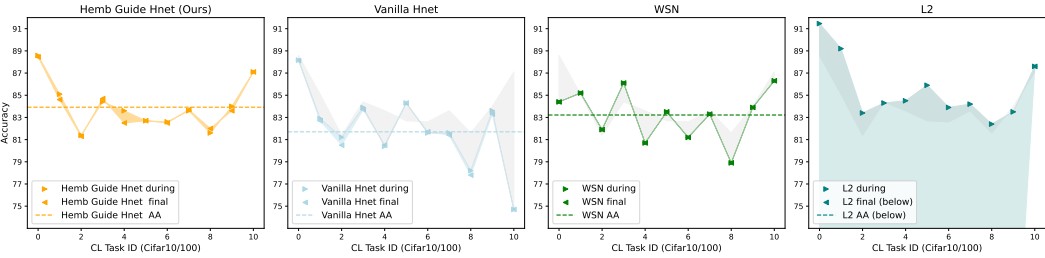

Figure 3: **The during and final task test accuracy of Cifar10/100 benchmark (ResNet-32 backbone, 100 epochs),** with axis x for CL task IDs and y for the test accuracy. In the figures, ▷ and ◁ denote the during and final accuracy, while the dashed line shows the average final accuracy and colored region represents their discrepancy, i.e. AA and BWT. From left to right is the accuracy visualization for H-embedding guided hypernet (ours), vanilla hypernet, WSN and L2 respectively. The grey regions in the right three figures denote the margin of during accuracy between these baselines and our method, i.e. discrepancy of FWT.

**Optimal Overall Transfer Ability** We select the best-performing baselines from each methodology category and plot their task-specific performance in Fig. 3. Each task is presented with two test accuracies: the accuracy obtained upon finishing training on the task, and the accuracy achieved by the final model after learning all CL tasks. As illustrated in the figures, our H-embedding guided hypernet demonstrates a notable advantage over vanilla hypernet and WSN, exhibiting both effectiveness and stability in forward transfer while performing comparably in backward transfer. On the

other hand, L2 as a regularization baseline, achieves good forward transfer ability, but fails in the mitigation of catastrophic forgetting. On the whole, our method displays a steady boost in forward transfer while retaining a competitive backward transfer, showcasing the best overall transfer ability, thereby attaining the highest average performance.

**Quicker Convergence**   With the intention of understanding how our guidance aids the training process, we visualize the test accuracy trend during the training stage of task 1, 4, 7, 11 of the 11 CL tasks under Cifar-ResNet setting in Fig. 4. It is shown in figures that, compared to a hypernet without H-embedding guidance, our method converges noticeably faster and achieves a higher final accuracy performance, especially with the growth of task numbers. Such phenomenon serves as a further suggestion that our H-embedding guidance provides substantial enhancement to the task learning in CL through forward transfer.

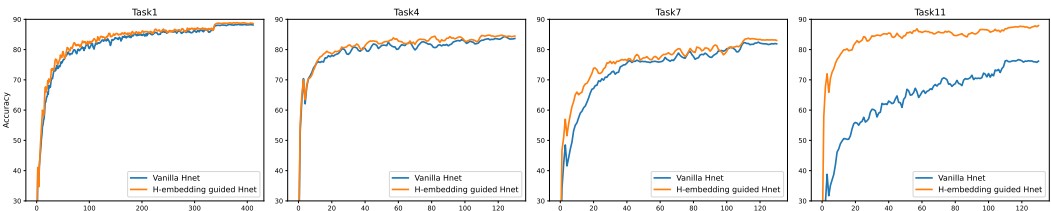

Figure 4: **Plotting of test accuracy during training task 1, 4, 7, 11 of Cifar10/100 benchmark,** with axis x and y for the number of checkpoints and accuracy respectively. The blue curve represents vanilla hypernet and orange represents our H-embedding guided hypernet. As CL progresses, our method exhibits quicker convergence to higher accuracy in later tasks.

**Embedding Interpretability**   To assess the task embeddings $\{e^{(j)}\}_{j=1}^{M}$ learned in our framework, we compute the task-wise Euclidean distances of the embeddings obtained with and without H-embedding guidance, and visualize the discrepancy between these two distance matrix in Fig. 5. The grid of $i$-th row and $j$-th column in the figure represents the relationship between task $i$ and task $j$. Darker cells indicate larger divergence between with- and without- guidance embeddings. Red signifies that the with-guidance embeddings result in a closer distance between the two tasks compared to the without-guidance embeddings, while blue represents the opposite. Take task 9, a Cifar100 split task covering classes of people and reptiles, as an instance. The embedding derived in our H-embedding guided hypernet successfully marks task 4, 6, 7, 8, 10 as more related, which all contain coverage of terrestrial animal classes or human scenarios. Our embedding also generally displays a greater

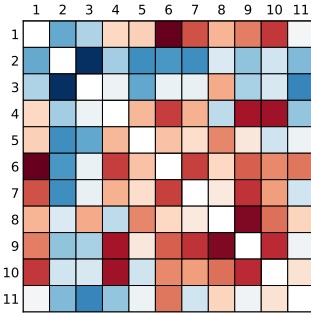

Figure 5: **Visualization of discrepancy between the task embedding distances learned w/ and w/o H-embedding guidance.**

preference on task 1, the more comprehensive Cifar10 task. Such correspondence with human intuition suggests a better capture of task interrelationships, leading to higher CL efficiency.

## 6   CONCLUSION

In this work, we propose a transferability task embedding guided hypernet to exploit the task relationships for continual learning. By introducing the information theoretical transferability based task embedding named H-embedding and incorporating it in a hypernetwork, we establish an online framework capable of capturing the statistical relations among the CL tasks and leveraging these knowledge for task-conditioned model weight guidance. Through extensive experimental studies, we validated that the adoption of H-embedding guidance enhances continual learning by facilitating forward transfer and improving the reliability of task embeddings, achieving the best final accuracy performance under various CL benchmarks.

## REPRODUCIBILITY STATEMENT

Experiments in our paper are run with fixed random seeds and are completely reproducible (with detailed information in Appendix. A.1). We also open-source the code implementation of our method at `https://anonymous.4open.science/r/H-embedding_guided_hypernet/` for better reproducibility and facilitating future researches.

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

# A APPENDIX

## A.1 EXPERIMENTAL SETTINGS

### A.1.1 COMPARISON EXPERIMENTS

**General Settings.** In CIFAR10/100 dataset using a ResNet-32 backbone network without pre-training, we evaluated several baseline methods include Finetune, Finetune-Head, EWC, L2, PredKD+FeatKD, and PackNet. The ResNet-32 uses 'option A', *i.e.*, leveraging the zero-padding shortcuts for increasing dimensions, as indicated in CIFAR-10 experiments of the original ResNet paper Sec4.2. All experiments were conducted using NVIDIA GeForce RTX 3090 GPUs.

To ensure a fair comparison, we adopted consistent training settings across all baseline methods except listing separately. Specifically, the batch size was set to 32, and each task was trained for 100

epochs. We used the Adam optimizer with an initial learning rate of 0.001. The learning rate was decayed by a factor of 10 after the 50th and 75th epochs. A weight decay of $1 \times 10^{-4}$ was applied. For robustness, each experiment was run three times with different random seeds 22, 32, and 42, and the results were averaged.

**Choice of Baselines.**  Our selection of baselines in this work aims to encompass a wide range of baseline categories, covering two of three primary categories in contemporary CL researches (*i.e.*, replay based, regularization based and architecture based), with the replay-based methods not conforming to our rehearsal-free setting. The specific choice of baselines in each category is mainly based on performance comparison conclusions in recent works such as Smith et al. (2023) and Kang et al. (2022). Therefore, we believe that our comparison study has included the most competitive and representative baselines.

**Details of Specific Methods.**  For our **H-embedding guided hypernet**, the learning rate of 0.0005 and the embedding loss beta is set to 0.05. For other **hypernet based baselines**, we primarily follows von Oswald et al. (2020) in the training settings with a learning rate of 0.001 and the CL loss beta being 0.05). The embedding loss beta is set to 0.05 (same as our H-embed hypernet) in Rand-embed hypernet. Notably, for all hypernet methods, we used exactly the same scheduling and transforming strategies as used by Oswald, and the embedding size is all set to 32.

For the **Finetune** baseline, the model was sequentially trained on each task without any mechanisms to prevent catastrophic forgetting. The model was randomly initialized and trained from scratch on the first task. For subsequent tasks, training continued using the weights obtained from the previous task.

In the **Finetune-Head** baseline, all convolutional layers of the ResNet-32 model were frozen after training on the first task. When learning new tasks, only the parameters of the final fully connected layer (the classifier) were updated. This approach aims to retain the feature representations learned from earlier tasks while adapting the classifier to new task-specific outputs.

For the **EWC** baseline (Kirkpatrick et al., 2017), we added a regularization term to the loss function to penalize significant changes to parameters important for previously learned tasks. The importance of each parameter was estimated using the Fisher Information Matrix. The regularization coefficient $\lambda$ was set to 10, following standard practice.

In the **L2** baseline, an L2 regularization term was added to the loss function to limit changes in the model parameters during training on new tasks. The regularization coefficient $\lambda$ was set to 1.0, determined by tuning on a small validation set derived from the training data of the first task.

For the **PredKD + FeatKD** method (Smith et al., 2023), we incorporated both prediction distillation and feature distillation to transfer knowledge from previous tasks to new ones. The distillation loss combines the Kullback-Leibler divergence between the soft outputs of the teacher (model trained on previous tasks) and the student (current model), as well as the mean squared error between their intermediate feature representations. The loss weights were set to $\alpha = 1.0$ and $\beta = 0.5$ based on preliminary tuning.

In the **PackNet** method (Mallya & Lazebnik, 2018), we employed iterative pruning to allocate dedicated network weights for each task. After training on each task, we pruned a certain percentage of the weights with the smallest magnitudes. Following the recommendations in the original paper, we experimented with pruning rates of 0.5, 0.75, and 0.8. We selected the pruning rate of 0.8, which yielded the best performance in our setting. After pruning, we fine-tuned the remaining weights for an additional 10 epochs with a reduced learning rate of $1 \times 10^{-4}$.

For the **WSN** method (Kang et al., 2022), we follows its original paper and use the default values of parameters in its official code repository. We choose the sparsity parameter $c = 0.5$ which performed best as listed in the WSN literature. Other parameters are set to the following values: optimization via Adam, a learning rate initialized at 1e-3 with a minimum of 1e-6, and a patience of 6 epochs for reducing the learning rate by a factor of 2. The models were trained for 100 epochs, with a batch size of 64 for both training and testing.

In all methods, we adhered to the principles of continual learning by not tuning hyperparameters on the full task set. Special care was taken in handling batch normalization layers, especially in meth-

ods involving parameter freezing or pruning. Following the settings in von Oswald et al. (2020), we stored and updated batch normalization statistics separately for each task to ensure proper normalization during both training and inference.

### A.1.2 Ablation Studies

**ImageNet-R** For the ImageNet-R dataset, we split the original 200 classes into ten 20-way classification tasks. Because of the uneven class sample size of ImageNet dataset, each task has varied numbers of training and test samples: Task 1 with 2,166 training samples and 543 test samples, Task 2 with 2,655 training and 716 test samples, ..., until Task 9 with 2,058 training samples and 471 test samples. In our method, we used a learning rate of 0.0005 and a the embedding loss beta of 0.05, training the models for 200 epochs. The backbone model is the same as used in comparison experiments. The results are derived on NVIDIA A800 GPUs, and is reproducible with random seed 42.

**Cifar10/100** The tasks in this setting are derived the same as in comparison studies. Yet, the backbone model is differently set to a 4-layer CNN as used by Zenke et al. (2017). We also followed Oswald in most of the hyperparameters, configuring learning rate to 0.0001, embedding size to 32, as well as using the same scheduling strategies. We trained each method with 100 epochs and the embedding loss beta is set to 0.2 for H-embed and rand-embed hypernets. The results are derived on NVIDIA GeForce RTX 3090 GPUs, and is reproducible with random seed 42.

**PermutedMNIST** Considering the smaller data dimension and model size in this setting, the embedding size is reduced to 24 and training iteration number is set to 5000. The backbone model on PermutedMNIST is selected to be an MLP with fully-connected layers of size 1000, 1000 as used by Van de Ven & Tolias (2019). We configured the learning rate as 0.0001 and the embedding loss beta as 0.05. The results are derived on NVIDIA GeForce RTX 3090 GPUs, and is reproducible with random seed 42.

### A.2 Detailed experimental results

Considering the limited space, we only presented the experimental results measured by our three metrics in main text. Here, we list the whole continual learning performance below. The during accuracy refers to the test accuracy of tasks upon finishing training on that task, and the final accuracy refers to the test accuracy of tasks when finishing learning all CL tasks. The results of comparison experiments are derived with three times running of seed 22, 32, 42 and the ablation studies are conducted with seed 42 only.

| Method | Task 1 | Task 2 | Task 3 | Task 4 | Task 5 | Task 6 | Task 7 | Task 8 | Task 9 | Task 10 | Task 11 |
|---|---|---|---|---|---|---|---|---|---|---|---|
| Finetune | 81.14 ± 0.32 | 79.10 ± 0.91 | 76.70 ± 0.85 | 79.97 ± 1.89 | 78.57 ± 2.40 | 79.97 ± 1.41 | 78.20 ± 1.92 | 79.70 ± 1.93 | 74.77 ± 1.46 | 80.43 ± 0.93 | 82.77 ± 2.08 |
| Finetune Head | 89.54 ± 0.44 | 75.80 ± 3.25 | 78.13 ± 1.65 | 79.67 ± 2.88 | 79.10 ± 0.85 | 78.47 ± 1.18 | 76.13 ± 0.80 | 80.83 ± 2.07 | 75.57 ± 0.38 | 81.77 ± 0.09 | 79.00 ± 0.42 |
| LwF | 88.66 ± 0.85 | 87.47 ± 0.47 | 83.87 ± 0.52 | 84.37 ± 0.94 | 84.17 ± 0.05 | 86.20 ± 0.71 | 82.77 ± 0.47 | 83.97 ± 0.52 | 82.63 ± 0.38 | 84.50 ± 1.27 | 86.87 ± 0.61 |
| EWC | 89.39 ± 1.64 | 86.57 ± 0.34 | 83.63 ± 1.14 | 86.10 ± 0.43 | 84.70 ± 1.36 | 85.67 ± 0.71 | 82.73 ± 1.23 | 82.30 ± 1.42 | 81.67 ± 0.91 | 83.13 ± 0.66 | 85.60 ± 1.92 |
| L2 | 91.27 ± 0.27 | 87.33 ± 1.34 | 84.07 ± 0.81 | 85.97 ± 1.28 | 85.53 ± 1.07 | 85.00 ± 0.83 | 84.20 ± 0.24 | 84.57 ± 0.33 | 81.27 ± 0.80 | 84.30 ± 0.86 | 87.33 ± 0.21 |
| PredKD+FeatKD | 88.66 ± 0.85 | 86.50 ± 0.28 | 82.70 ± 0.71 | 86.03 ± 0.47 | 85.80 ± 0.99 | 86.07 ± 0.24 | 83.60 ± 0.00 | 81.17 ± 0.52 | 81.10 ± 0.99 | 83.63 ± 0.09 | 85.43 ± 1.79 |
| Packnet | 82.79 ± 0.20 | 73.07 ± 0.63 | 69.03 ± 0.39 | 76.20 ± 0.78 | 69.47 ± 1.35 | 71.40 ± 0.99 | 68.07 ± 1.69 | 71.43 ± 0.62 | 67.13 ± 0.47 | 66.47 ± 0.34 | 74.50 ± 0.54 |
| WSN | 84.03 ± 0.26 | 83.63 ± 1.13 | 80.50 ± 1.10 | 84.80 ± 0.93 | 81.30 ± 0.45 | 83.53 ± 0.21 | 80.27 ± 0.68 | 83.27 ± 0.29 | 78.77 ± 1.55 | 84.87 ± 0.82 | 86.57 ± 0.60 |
| Vanilla Hnet | 88.52 ± 0.37 | 82.87 ± 0.29 | 80.47 ± 0.52 | 83.30 ± 0.43 | 81.97 ± 1.17 | 83.30 ± 0.78 | 80.37 ± 1.34 | 80.60 ± 2.79 | 79.13 ± 0.93 | 82.00 ± 1.35 | 82.30 ± 5.45 |
| Rand-embed Hnet | 88.64 ± 0.29 | 83.23 ± 0.61 | 80.77 ± 0.24 | 83.10 ± 0.86 | 82.30 ± 0.08 | 81.70 ± 1.06 | 80.10 ± 1.07 | 80.43 ± 1.91 | 78.83 ± 1.80 | 81.50 ± 0.14 | 86.00 ± 1.22 |
| **H-embed Hnet*** | 88.67 ± 0.42 | 82.77 ± 1.77 | 81.13 ± 0.21 | 84.43 ± 0.53 | 83.40 ± 0.16 | 83.07 ± 0.45 | 82.10 ± 1.00 | 82.43 ± 0.98 | 80.47 ± 1.03 | 83.93 ± 0.49 | 86.87 ± 0.40 |

Table 4: **Comparison Experiments, Accuracy During.** The test accuracy of tasks upon finishing training on that task. The mean value and standard deviation are derived with three times running.

| Method | Task 1 | Task 2 | Task 3 | Task 4 | Task 5 | Task 6 | Task 7 | Task 8 | Task 9 | Task 10 | Task 11 |
|---|---|---|---|---|---|---|---|---|---|---|---|
| Finetune | 10.21 ± 0.30 | 10.27 ± 0.77 | 10.00 ± 0.00 | 9.63 ± 0.52 | 11.80 ± 1.81 | 11.83 ± 2.66 | 11.07 ± 2.66 | 10.90 ± 1.65 | 13.23 ± 3.44 | 19.77 ± 1.19 | 82.77 ± 2.08 |
| Finetune Head | 9.48 ± 0.76 | 7.97 ± 0.94 | 8.70 ± 0.28 | 8.13 ± 0.24 | 9.80 ± 0.28 | 6.57 ± 2.03 | 11.07 ± 0.66 | 9.80 ± 1.84 | 9.73 ± 0.24 | 8.60 ± 0.99 | 79.00 ± 0.42 |
| LwF | 15.88 ± 0.95 | 19.63 ± 1.37 | 21.17 ± 1.04 | 18.43 ± 0.38 | 22.07 ± 0.66 | 22.30 ± 1.84 | 26.67 ± 1.89 | 36.23 ± 4.48 | 40.77 ± 4.43 | 53.20 ± 0.57 | 86.87 ± 0.61 |
| EWC | 19.34 ± 4.27 | 18.00 ± 3.76 | 23.23 ± 3.93 | 25.10 ± 3.77 | 24.97 ± 2.77 | 30.00 ± 6.34 | 31.67 ± 3.60 | 40.87 ± 2.17 | 41.03 ± 7.99 | 57.80 ± 5.03 | 85.60 ± 1.92 |
| L2 | 23.32 ± 5.53 | 21.60 ± 3.27 | 25.10 ± 0.86 | 25.17 ± 5.00 | 29.30 ± 2.79 | 27.83 ± 5.42 | 41.07 ± 3.13 | 49.30 ± 6.23 | 51.23 ± 2.01 | 56.93 ± 1.61 | 87.33 ± 0.21 |
| PredKD+FeatKD | 19.12 ± 5.04 | 15.53 ± 2.29 | 23.37 ± 1.69 | 20.80 ± 1.27 | 19.93 ± 2.15 | 25.23 ± 4.86 | 23.90 ± 0.41 | 32.93 ± 5.14 | 43.77 ± 3.30 | 51.73 ± 4.43 | 87.07 ± 0.17 |
| Packnet | 82.79 ± 0.20 | 73.07 ± 0.63 | 69.03 ± 0.39 | 76.20 ± 0.78 | 69.47 ± 1.35 | 71.40 ± 0.99 | 68.07 ± 1.69 | 71.43 ± 0.62 | 67.13 ± 0.47 | 66.47 ± 0.34 | 74.50 ± 0.54 |
| WSN | 84.03 ± 0.26 | 83.63 ± 1.13 | 80.50 ± 1.10 | 84.80 ± 0.93 | 81.30 ± 0.45 | 83.53 ± 0.21 | 80.27 ± 0.68 | 83.27 ± 0.29 | 78.77 ± 1.55 | 84.87 ± 0.82 | 86.57 ± 0.60 |
| Vanilla Hnet | 88.55 ± 0.37 | 82.73 ± 0.37 | 80.13 ± 0.33 | 83.40 ± 0.24 | 82.00 ± 1.07 | 83.27 ± 0.82 | 80.43 ± 1.58 | 80.53 ± 2.60 | 78.97 ± 1.08 | 81.97 ± 1.20 | 82.30 ± 5.45 |
| Rand-embed Hnet | 88.64 ± 0.30 | 83.27 ± 0.56 | 80.80 ± 0.24 | 83.40 ± 0.59 | 82.27 ± 0.12 | 81.70 ± 1.06 | 79.93 ± 1.08 | 80.30 ± 1.76 | 78.67 ± 1.91 | 81.67 ± 0.09 | 86.00 ± 1.22 |
| **H-embed Hnet*** | 88.61 ± 0.46 | 82.50 ± 1.64 | 81.17 ± 0.12 | 84.63 ± 0.65 | 83.10 ± 0.45 | 83.03 ± 0.34 | 81.87 ± 1.04 | 82.57 ± 1.09 | 80.50 ± 1.19 | 83.80 ± 0.43 | 86.87 ± 0.40 |

Table 5: **Comparison Experiments, Accuracy Final.** The test accuracy of tasks when finishing learning all CL tasks. The mean value and standard deviation are derived with three times running.

| Setting | Method | Type | Task 1 | Task 2 | Task 3 | Task 4 | Task 5 | Task 6 | Task 7 | Task 8 | Task 9 | Task 10 | Task 11 |
|---|---|---|---|---|---|---|---|---|---|---|---|---|---|
| **PermutedMNIST** | Rand-embed | Final | 98.09 | 97.83 | 97.58 | 97.85 | 97.83 | 97.48 | 97.08 | 97.17 | 96.86 | 96.87 | |
| | | During | 98.07 | 97.79 | 97.62 | 97.85 | 97.85 | 97.46 | 97.09 | 97.05 | 96.82 | 96.87 | |
| | H-embed | Final | 98.04 | 97.82 | 97.49 | 97.93 | 97.91 | 97.71 | 97.53 | 97.39 | 97.10 | 96.78 | |
| | | During | 98.07 | 97.79 | 97.48 | 97.99 | 97.92 | 97.75 | 97.51 | 97.40 | 97.13 | 96.78 | |
| | Vanilla | Final | 98.10 | 97.78 | 97.79 | 97.50 | 97.60 | 97.46 | 97.37 | 97.17 | 97.21 | 96.97 | |
| | | During | 98.07 | 97.79 | 97.83 | 97.51 | 97.55 | 97.46 | 97.38 | 97.14 | 97.19 | 96.97 | |
| **Cifar10/100** | Rand-embed | Final | 70.47 | 71.80 | 69.10 | 71.40 | 64.60 | 69.10 | 66.70 | 72.80 | 71.30 | 74.50 | 81.20 |
| | | During | 79.47 | 76.60 | 71.90 | 78.60 | 75.00 | 77.00 | 76.20 | 76.00 | 75.00 | 77.40 | 81.20 |
| | H-embed | Final | 72.35 | 68.60 | 66.80 | 72.10 | 65.50 | 69.70 | 69.00 | 73.80 | 72.90 | 75.40 | 83.30 |
| | | During | 79.15 | 76.90 | 73.40 | 79.20 | 76.30 | 78.30 | 76.70 | 76.50 | 75.20 | 78.90 | 83.30 |
| | Vanilla | Final | 70.27 | 71.70 | 65.90 | 69.90 | 65.70 | 67.30 | 64.30 | 70.30 | 69.70 | 71.00 | 80.40 |
| | | During | 79.47 | 76.60 | 71.70 | 78.00 | 75.80 | 77.50 | 76.50 | 76.20 | 75.70 | 76.50 | 80.40 |
| **ImageNet-R** | Rand-embed | Final | 41.25 | 31.84 | 42.68 | 33.97 | 41.61 | 38.35 | 35.73 | 38.68 | 38.84 | 45.65 | |
| | | During | 41.99 | 32.40 | 43.40 | 33.53 | 42.10 | 37.56 | 35.73 | 38.99 | 39.18 | 45.65 | |
| | H-embed | Final | 41.80 | 32.40 | 40.33 | 35.86 | 44.84 | 39.14 | 39.50 | 39.62 | 36.97 | 45.01 | |
| | | During | 41.99 | 32.40 | 40.87 | 36.30 | 44.68 | 39.78 | 39.14 | 39.47 | 36.80 | 45.01 | |
| | Vanilla | Final | 41.07 | 32.12 | 38.88 | 33.67 | 46.61 | 37.08 | 36.62 | 39.31 | 36.80 | 42.68 | |
| | | During | 41.99 | 32.40 | 39.78 | 33.97 | 46.29 | 37.24 | 36.80 | 39.15 | 36.80 | 42.68 | |

Table 6: **Ablation Studies, During and Final Accuracy.**