# OpenReview forum: "Exploiting Task Relationships for Continual Learning with Transferability-aware Task Embedding"
_ICLR.cc/2025/Conference — ICLR 2025 Conference Withdrawn Submission_

### Official Review · Reviewer_MLMK · 2024-10-29

**Soundness:** 3
**Presentation:** 2
**Contribution:** 1
**Rating:** 3
**Confidence:** 5

**Summary:**

This paper considers the task relationships in continual learning and introduces an online H-embedding scheme based on the concept of H-score from information theory. By applying additional constraints to the update of task embeddings during the learning process, the method better captures task relationships. The approach is based on a hypernetwork, ensuring model scalability; specifically, computational and storage overheads do not increase significantly as the task sequence grows. Additionally, the H-embedding design promotes positive forward transfer, enhancing hypernetwork performance in task-incremental learning and outperforming comparative methods.

**Strengths:**

1. The motivation to consider the task relationships in continual learning is sound, and designing a method to improve task embedding learning in a hypernetwork is indeed valuable.
2. It is crucial to consider both backward and forward transfer in evaluating continual learning. Through its experimental setup, the paper demonstrates that H-embedding effectively promotes forward knowledge transfer.
3. The proposed method is efficient: as the task sequence lengthens, the computational and storage costs remain relatively stable, which adds value for real-world applications.

**Weaknesses:**

1. The experiments are limited to task-incremental learning, while in real-world scenarios, task identifiers are often unavailable during the inference stage. This limitation may significantly restrict the method's contribution, as it cannot be applied in the more challenging class-incremental learning setting.
2. The selected datasets are not diverse enough; the accuracy on PermutedMNIST is nearing saturation, and CIFAR-10/100 are relatively low in difficulty. The authors should consider more varied datasets, such as Split DomainNet, to strengthen the persuasiveness of the results.
3. The core contribution of this work mainly involves adding constraints to the task embedding learning process, which does not significantly diverge from the vanilla hypernetwork approach. Additionally, the performance improvement over the original hypernetwork is not very pronounced.
4. The experiments lack results on more advanced model architectures, such as ViT, and comparisons with continual learning methods based on pretrained models.
5. The writing quality is poor, with multiple grammatical errors, for example, the sentences in L188-191 are not correct.

**Questions:**

1. Rand-embed Hnet is an important baseline in this paper, but there is no detailed explanation of its setup. Could the authors clarify the settings of this baseline?
2. When performing task-incremental learning, was a multi-head design adopted, i.e., was a separate classification head set for each task?
3. Can the proposed method also be applied to class-incremental learning tasks?

---

### Official Review · Reviewer_MRPj · 2024-10-31

**Soundness:** 3
**Presentation:** 2
**Contribution:** 2
**Rating:** 6
**Confidence:** 5

**Summary:**

This paper introduces a novel transferability-based task embedding, called H-embedding, incorporated within a hypernetwork to enhance continual learning (CL) by leveraging inter-task relationships. This approach aims to capture statistical relations among tasks, enabling task-conditioned model weights that mitigate catastrophic forgetting.

**Strengths:**

1. It is an essential problem to focus to learn more about inter-task relationships to aid to CL.
2. Task-conditioned hyper net is a smart strategy to tackle the problem, a little more background in hyper nets would have been helpful.
3. Priors are well incorporated with the H-embedding proposed
4. Extensive performance study beyond just accuracy strengths the contributions.

**Weaknesses:**

1. Doesn't forward and backward transfer highlight these relationships? How does your method differentiate?
2. Although the authors have considered more generic settings, it is essential to compare in realistic scenarios as defined in [1,2].
3. Need to compare with more recent transformer backbone methods and other SoTA methods, existing experiments do not seem fair.



[1] Online Class-Incremental Learning For Real-World Food Image Classification - Siddeshwar Raghavan, Jiangpeng He, Fengqing Zhu
[2] Generalized Class Incremental Learning - Fei Mi; Lingjing Kong; Tao Lin; Kaicheng Yu; Boi Faltings

**Questions:**

Please refer to the weakness.

---

### Official Review · Reviewer_4JEF · 2024-11-03

**Soundness:** 3
**Presentation:** 4
**Contribution:** 3
**Rating:** 5
**Confidence:** 4

**Summary:**

The authors present a work that introduces a new way to solve continual learning problems. By using a single hypernetwork and task embeddings for each task, it can produce weights for the model corresponding to each task. To prevent hypernetwork itself forgetting, it acquire several loss functions to ensure it will generate similar weight with same inputs after training several new tasks.

**Strengths:**

1. Good performance gain on several datasets compared to previous works.
2. Well-written and detailed descriptions in most sections. Even the details of other methods are clearly provided in the appendix. Solid experimental results.
3. Providing code is a plus.
4. Using H-score to guide the model sounds solid and works well.

**Weaknesses:**

Listed at questions. Generally a good work, but I would like the authors show the robustness and efficiency of the proposed methods.

**Questions:**

1. The method does not extend to class-incremental learning for inference, which may not be entirely orthogonal to the authors' approach. Since task embeddings are generated during training, it might be possible to exploit these embeddings to identify classes during inference. I would like to see if the authors can extend to the more challenging class-incremental settings or explian why it is not applicable.
2. How stable is the hypernetwork itself? Will it degrade after a long run? Will it be hard to tune the hypernetwork? I belive it is crucial to present that the proposed methods is robust enough.
3. Although the embedding is small, will training time linearly increase with the number of previous tasks?
4. The use of the hypernet to generate weights for each task is not entirely concise, in my opinion. How long does it take to generate weights during inference? What if the incoming task switches frequently? I would like to know the inference speed when dealing with an extreme case where tasks switch with each iteration compared to a standard model.

---

### Official Review · Reviewer_F4ZV · 2024-11-07

**Soundness:** 2
**Presentation:** 3
**Contribution:** 2
**Rating:** 5
**Confidence:** 4

**Summary:**

This paper proposes a novel hypernet framework, where the learning of task embedding is guided by the continual task relationships. The method introduces a novel form of embedding called H-embedding, utilizing a encoder-decoder architecture to integrate into hypernet. This approach is based on information theory and provides a new theoretical perspective for the field of continuous learning.

**Strengths:**

1. The paper focuses on continual learning with hypernet, which is a highly valuable topic. By designing H-embedding, it can help hypernet to effectively capture task relationships.
2. The paper provides a interesting visualization between task embeddings, promoting the interpretability of the method.
3. The paper provides a detailed explanation of the methodology, which enhance the reproducibility of the research.

**Weaknesses:**

1. H-embedding is based on the assumption that there is a correlation between tasks. If the differences between tasks are very large, this assumption may not hold, thus affecting model performance.
2. The benchmarks conducted for the main experiment are few and limited, the experiment setting(Task increment) is also trival, which can not fully demonstrate the performance of the method.
3. The main experiment section lacks comparison with the latest methods, such as DAP[1].
4. The ablation study shows that the introduction of H-embedding is only marginal improvements.
[1] Jung D, Han D, Bang J, et al. Generating instance-level prompts for rehearsal-free continual learning[C]//Proceedings of the IEEE/CVF International Conference on Computer Vision. 2023: 11847-11857.

**Questions:**

1. I am curious about the performance of H-embedding in the scenario where the data distribution of between tasks varies significantly, such as DomainNet.
2. Due to the assumption that there is a correlation between tasks, whether the performance of H-embedding will be affected when the task number is extremely large(e.g. 69 in DomainNet).

---

### Author Response · Authors · 2024-11-19

We are grateful to all reviewers of their treasureable suggestions and critics of our paper. However, as we recognize the existing limitations of the current version, we decide to withdraw the paper and polish it up for future submissions. Thanks again to all the feedbacks and we are open to any further discussions.

---

### Note · Authors · 2024-11-19

I have read and agree with the venue's withdrawal policy on behalf of myself and my co-authors.